# Utilizing Physics-Based Input Features within a Machine Learning Model to Predict Wind Speed Forecasting Error

Daniel Vassallo[1], Raghavendra Krishnamurthy[2, 1], and Harindra J.S. Fernando[1]

[1]University of Notre Dame, Indiana, USA
[2]Pacific Northwest National Laboratory, Washington, USA

**Correspondence:** Daniel Vassallo (dvassall@nd.edu)

**Abstract.** Machine learning is quickly becoming a commonly used technique for wind speed and power forecasting. Many of these methods utilize exogenous variables as input features, but there remains the question of which atmospheric variables provide the most predictive power, especially in handling non-linearities that lead to forecasting error. This investigation addresses this question via creation of a hybrid model that utilizes an autoregressive integrated moving average (ARIMA) model to make an initial wind speed forecast followed by a random forest model that attempts to predict the ARIMA forecasting error using knowledge of exogenous atmospheric variables. Variables conveying information about atmospheric stability and turbulence as well as inertial forcing are found to be useful in dealing with non-linear error prediction. Wind direction and temperature are found to be the most beneficial individual input features. Streamwise wind speed, time of day, turbulence intensity, turbulent heat flux, wind direction, and temperature are found to be particularly useful when used in unison. The prediction accuracy of the ARIMA-RF hybrid is compared to that of the persistence and bias-corrected ARIMA models. The ARIMA-RF model is shown to improve upon the latter commonly employed modeling methods, reducing hourly forecasting error by approximately 30% below that of the bias-corrected ARIMA model.

## 1 Introduction

Global wind power capacity reached almost 600 GW at the end of 2018 (GWEC, 2019), making wind energy a vital component of international electricity markets. Unfortunately, integrating wind power into an existing electrical grid is difficult because of wind resource intermittency and forecasting complexity. For utility companies employing wind power, it is important to estimate the aggregated load over a period of time to better balance grid resources, including long-term (1+ days ahead), short-term (1-3 hours ahead) and very-short term (15 minutes ahead) forecasts (Soman et al., 2010; Wu et al., 2012). Forecasting accuracy depends on site conditions, surrounding terrain, and local meteorology. Many wind farms are built in locations which are known to amplify winds due to surrounding terrain (such as Lake Turkana in Kenya, Tehachapi Pass in California etc.), requiring bespoke forecasts for accurate predictions. Numerical weather prediction models (NWPs) fail at such complex sites due to a lack of appropriate parameterization schemes suitable for local conditions (Akish et al., 2019; Bianco et al., 2019; Olson et al., 2019; Stiperski et al., 2019). Therefore, statistical models and computational learning systems (such as an artificial neural network or random forest) are likely better suited to provide accurate power forecasts. Since wind power production is

heavily reliant upon environmental conditions, improvements in wind speed forecasting would allow for more reliable wind power forecasts.

If we simplify our wind speed prediction process down to its core (which has no true relation to atmospheric motions), we can imagine a system of atmospheric flow without external forcing. This would result in a constant streamwise wind speed $U$ (i.e. $U_\tau = U_{\tau-1}$; $U$ is streamwise wind speed, $\tau$ a timestep; this assumes discrete timesteps for simplicity). In this case, a persistence or autoregressive forecast would have zero forecasting error and uncertainty. However, uncertainty increases once we add an external force that we may represent by some variable $x_1$. Now future wind speed may be seen to be $U_\tau = f(U_{\tau-1}, x_{1,\tau-1})$. Assuming the external force is notable in strength and coupled with the inertia associated with winds, the previous autoregressive model will now struggle to predict $U_\tau$ because it does not take into account our external forcing $x_{1,\tau-1}$, resulting in an error $\varepsilon$ ($\varepsilon_\tau$ is abbreviated to $\varepsilon$ for simplicity). We can then break down our future wind speed into two parts: $U_\tau = \hat{U}_\tau + \varepsilon$ where $\hat{U}_\tau$ is our autoregressive forecast that is only dependent on $U_{\tau-1}$ (i.e. $\hat{U}_\tau = f(U_{\tau-1})$). The prediction error is thus skewed to represent the effects of the external force $x_{1,\tau-1}$ upon $U_{\tau-1}$.

If we continue to add external forces ($x_1$, $x_2$, ... $x_n$; $n$ is the number of external forcing variables), our atmospheric system becomes much more complex and non-linear due to interactions between forcing mechanisms. We can again obtain our forecasting error as $\varepsilon = f(U_{\tau-1}, x_{1,\tau-1}, x_{2,\tau-1}, ... x_{n,\tau-1})$, which we can discretize as $\varepsilon = \mu_\varepsilon + \varepsilon'$ ($\mu_\varepsilon$ is the error bias, $\varepsilon'$ the error fluctuations about $\mu_\varepsilon$) given that we have a statistically significant sample size and the process is stationary. Squaring this equation and taking the average gives us the discretized equation for the mean squared error $\overline{\varepsilon^2} = \overline{\mu_\varepsilon^2} + \overline{\varepsilon'^2}$, with $\overline{\varepsilon'^2}$ representing the error variance and overlines denoting the average over all samples (Lange, 2005). $\overline{\mu_\varepsilon^2}$ represents the bias and may be removed via a simple bias-correction. The true concern is the error fluctuation term ($\varepsilon'$) which constitutes the error variance. Assuming the external forcing variables ($x$'s) are normally distributed, we can break down $\overline{\varepsilon'^2}$ into two constituents (Ku et al., 1966):

$$\overline{\varepsilon'^2} = \sigma_{x_j}^2 \left( \frac{\partial \varepsilon}{\partial x_j} \right)^2 + 2 \left[ \sigma_{x_j, x_k} \frac{\partial \varepsilon}{\partial x_j} \frac{\partial \varepsilon}{\partial x_k} \right], \qquad j \neq k \tag{1}$$

where $\sigma_{x_j}^2$ is the variance of $x_j$ and $\sigma_{x_j, x_k}$ is the co-variance between $x_j$ and $x_k$ (subscript $\tau$ removed for simplicity). Unless external forcing (or its coupling with $U_{\tau-1}$) is minimal, the error is likely highly non-linear and chaotic (i.e. large $\overline{\varepsilon'^2}$). Therefore, it behooves us to discover which forcing mechanisms and atmospheric variables are the best predictors of individual fluctuations $\varepsilon'$, which we will call "exogenous error".

Many studies that use machine learning (ML) techniques for wind speed or power forecasting utilize a handful of unadulterated atmospheric variables such as wind speed, pressure, and temperature as input features (Mohandes et al., 2004; Ramasamy et al., 2015; Lazarevska, 2018; Chen et al., 2019). Recently, a handful of investigations have begun to determine which variables may be most useful for these models. Vassallo et al. (2020a) showed that invoking turbulence intensity ($TI$) can vastly improve vertical wind speed extrapolation accuracy. Similarly, Li et al. (2019) showed that $TI$ improves wind speed forecasting on multiple timescales, while Optis and Perr-Sauer (2019) showed that both atmospheric stability and turbulence levels are important indicators for wind power forecasting. Markedly, it has been shown by Cadenas et al. (2016) that multivariate sta-

tistical models consistently outperform univariate models for wind speed forecasting. However, to the authors' knowledge, the

question of which atmospheric variables are most useful in predicting exogenous error has not been addressed in the literature.

This investigation aims to determine if exogenous error may be, at least in part, predicted via a list of common meteorological measurements by following a methodology similar to that performed by Cadenas and Rivera (2010). The autoregressive integrated moving average (ARIMA) model first obtains an autoregressive forecast, and the forecasting error is extracted and bias-corrected. A random forest model is then utilized to discover patterns in the exogenous variables (and their relations to

the endogenous variable $U$) that are predictive of exogenous error. The ARIMA-random forest hybrid model so constructed is referred to as the ARIMA-RF model.

This study is not intended to provide a catch-all list of input features that should or should not be used for every future study. Rather, it aims to inform future researchers and industry professionals as to what types of meteorological information must be used as ML inputs to predict the non-linear interactions between various atmospheric forces. Section 2 describes

the Perdigão field campaign (the data source for the work), site characteristics, and instrumentation used for data collection. Section 3 provides an overview of the models utilized, testing process, and feature extraction/selection methodology. Section 4 provides testing results and Section 5 includes a brief discussion of the obtained results. Finally, conclusions can be found in Section 6.

## 2   Site, Data, & Instrumentation

Data for this study were taken from the Perdigão campaign, a multinational project located in central Portugal that took place in the spring of 2017 (Fernando et al., 2019). The project site is characterized by two parallel ridges, both about 5 km in length with a 1.5 km wide valley between them. These ridges, which are represented by the elevated contours in Fig. 1, run northwest to southeast and rise about 250 m above the surrounding topography, making the site highly complex and increasing forecasting difficulty. The ridges will be referred to as the northern and southern ridge.

A variety of remote and *in situ* sensors were positioned in and around the valley to provide an accurate and thorough description of the surrounding flow field. Foremost among these sensors was a grid of meteorological towers which ran both parallel and normal to the ridges. One 100 m tower located on top of the northern ridge (white star in Fig. 1) is utilized in this study. This tower had sonic anemometers (20 Hz native measurement resolution) at 10, 20, 30, 40, 60, 80, and 100 m above ground level (AGL) as well as temperature sensors at 2, 10, 20, 40, 60, 80, and 100 m AGL. Information about tower data

quality control, including corrections for boom orientation and tilt, may be found in NCAR/UCAR (2019). No clear tower wake effects could be discerned. The tower data in the Perdigão database has been averaged into 5-minute increments by data managers at NCAR.

Sensors at 20 and 100 m AGL were chosen because of the high percentage ($> 99\%$ for all variables except temperature at 100 m AGL, which was available for $\sim 95\%$ of the periods) of clean data at these elevations. The utilized data spans three

90   months, running from 10 March – 16 June 2017. Data at 100 m were correlated with that at 20 m, and missing data were filled using the variance ratio measure-correlate-predict method (Rogers et al., 2005). Any periods unavailable at both heights were

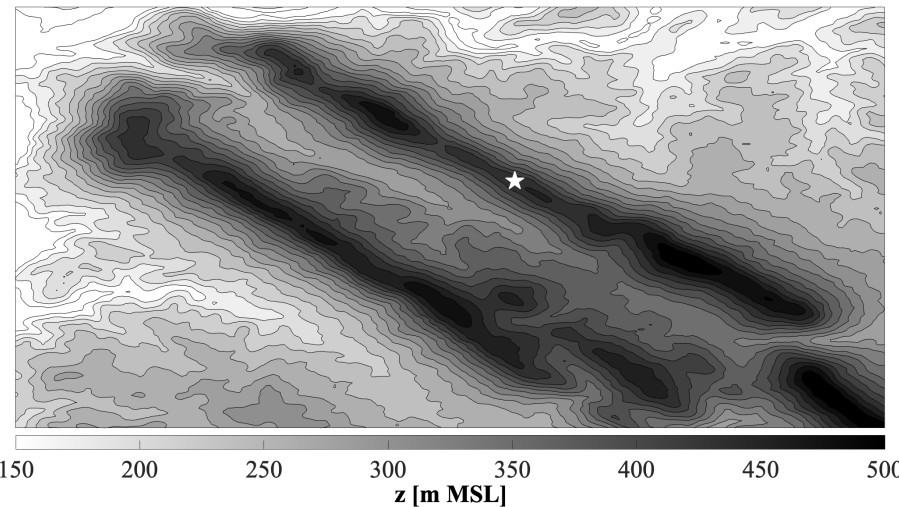

**Figure 1.** Contour plot of the campaign topography in meters above mean sea level (MSL). The white star represents the 100 m tower location on the northern ridge.

filled using linear interpolation with Gaussian noise. All periods are required for proper functionality and assessment of the ARIMA model, and manually filled periods are not expected to make a noticeable difference in the findings.

The augmented data were averaged into 10-minute, hourly, and three-hour segments at a 5-minute moving average in order
to create a robust dataset (over 28,000 samples). These three datasets (representing the same information at different averaging intervals) were then randomly split into training (75% of the input-output pairs) and testing (the final 25% of input-output pairs) sets. To ease concerns of the model overfitting the overlapping dataset, each internal node in the random forest model (which already has built-in mechanisms that severely hinder overfitting, as described by Breiman (2001) and James et al. (2013); model described in Section 3.2) was required to contain at least 100 samples in order to split (i.e. each branch of every decision tree
stops splitting once there are less than 100 samples).

The target streamwise wind speed, or that to be forecasted, is located at 100 m AGL. Squared buoyancy frequency ($N^2$), Richardson numbers (flux $Ri_f$ and gradient $Ri_g$), and temperature gradient ($\partial T/\partial z$) were calculated between 20 – 100 m AGL. Friction velocity ($u^*$) was found at 20 m, just above surface roughness height (Fernando et al., 2019). All other input variables utilized were from 100 m AGL.

**3  Methodology**

This investigation utilizes two modeling methods, ARIMA and random forest regression, to create a hybrid model (ARIMA-RF) wherein the ARIMA model is first used to get a linear, univariate wind speed forecast. The ARIMA forecast is bias-corrected and the exogenous error is then extracted and used as the target variable for the random forest. The random forest's

goal (and the goal of the study) is to determine which atmospheric variables and forcing categories are useful for the prediction of exogenous error. After the most important individual variables have been established, combinations of these input features are tested in an effort to determine whether multiple variables and/or informational categories can be coupled to improve exogenous error prediction. Finally, the ARIMA-RF results are compared with those of the persistence method and bias-corrected ARIMA model. 75% of the samples (input-output pairs representing the training set) are randomly selected and used for model construction and bias calculation. The final 25% of the samples are set aside for testing to enable a direct, blind comparison between all models. Section 3.1 details the ARIMA model, while Section 3.2 describes random forest regression. Sections 3.3 and 3.4 provide more detail on the feature extraction and selection methodology as well as the testing procedure.

## 3.1 ARIMA

ARIMA (Box et al., 2015) is a univariate statistical model used for time series forecasting. It is predicated on the combination of three functions: an autoregressive function that uses lagged values as inputs, a moving average function that uses past forecasting errors as inputs, and a differencing function used to make a time series stationary. In its simplest form, the next term in a time series sequence, $y_\tau$, is given by

$$y_\tau = \sum_{i=1}^{p} \phi_i y_{\tau-i} + \sum_{j=1}^{q} \Theta_j \varepsilon_{\tau-j} + \varepsilon_\tau \tag{2}$$

where $p$ and $q$ are the orders of the autoregressive and moving average functions, respectively, $\phi_i$ and $\Theta_j$ the $i^{th}$ autoregressive and $j^{th}$ moving average parameters, respectively, $y_{\tau-i}$ the $i^{th}$ lagged value, $\varepsilon_{\tau-j}$ the $j^{th}$ past prediction error, and $\varepsilon_\tau$ the error term at time $\tau$. The order of differencing is given by the parameter $d$ and does not show up directly in Eqn. 2.

The dataset was tested for long-term statistical stationarity via the Augmented Dickey Fuller Test (Dickey and Fuller, 1979) using the statsmodels Python module (Seabold and Perktold, 2010). The test, to a statistically significant degree, proved that the wind speed data contains no embedded trends or drift (e.g. changes in the mean or variance of the wind speed due to long-term variability). Therefore, the differencing parameter $d$ was set to 0 (This turns the ARIMA model into an ARMA model, but we stick with the term ARIMA for uniformity). The autoregressive and moving average parameters used, $p = 2$ and $q = 1$, were determined via minimization of the Akaike information criterion (Shibata, 1976) and empirical testing. Increasing parameters beyond this point did not lead to improved ARIMA accuracy. Although the wind speed data is stationary, general atmospheric seasonality (Chervin, 1986; Ramana et al., 2004) is expected to have an impact on multiple input features, requiring training and testing data to be randomly shuffled.

## 3.2 Random Forest Regression

Random forest regression (Breiman, 2001) is an ensemble method that is made up of a population of decision trees. Bootstrap aggregation (bagging) is used so that each tree can randomly sample from the dataset with replacement, while only a random subset of the total feature set is given to each individual tree. The trees can be pruned (truncated) to add further diversification.

After construction, the population's individual predictions are averaged to give a final prediction of the target variable. Ideally, this process results in a diversified and decorrelated set of trees whose predictive errors cancel out, producing a more robust final prediction.

An advantage of random forests is their ability to determine the importance of all input features for the predictive process. This is done by calculating the mean decrease impurity, or the decrease in variance that is achieved during a given split in each decision tree. The decrease in impurity for each input feature can be averaged over the entire forest, providing an approximation of the feature's importance for the prediction (feature importance estimates sum to 100% to ease interpretability). However, if two input variables are highly correlated (as is expected when testing atmospheric forcing), it is highly unlikely that the reported values will accurately represent each variable's significance (Breiman, 2001). Therefore, each variable is first tested individually to determine its individual benefits prior to coupling with other exogenous variables. To assist the random forest in representing the dynamic nature of atmospheric processes, input variables are taken from the previous two timesteps (i.e. input feature $U$ comprises $U_{\tau-1}$ and $U_{\tau-2}$).

The constructed random forest model contains 1,000 trees for tests of individual variables and 1,500 trees for tests of variable combinations. This was found to be sufficiently large to ensure prediction stability (to within a root mean square error of $\pm 0.001$ m s$^{-1}$), and the inclusion of additional trees does not result in higher prediction accuracy. To ease concerns of overfitting, each internal node was required have at least 100 samples in order to split (this truncation is a form of regularization). The random forest model was built using the scikit-learn Python library (Pedregosa et al., 2011).

## 3.3 Feature Extraction and Selection

In an effort to ensure that the findings are applicable to real-world campaigns, we limit our sources of information to those which may be measured by a typical meteorological mast containing sonic anemometers alongside temperature sensors. Using this information, we can write our future wind speed $U_\tau$ as a function of the following variables, which were broken down into their mean and fluctuating values:

$$U_\tau = f\left(U_i, \theta_i, W_i, T_i, t_i, u_i', \theta_i', w_i', T_i'\right) \tag{3}$$

where $U_i$ and $\theta_i$ are the mean streamwise wind speed and direction, respectively, $W_i$ the mean vertical wind speed, $T_i$ the mean temperature, $t_i$ the time of day, $u_i'$ the fluctuating horizontal velocity, $\theta_i'$ the fluctuating wind direction, $w_i'$ the fluctuating vertical velocity, and $T_i'$ the fluctuating temperature at each previous timestep $i$. Unfortunately, $\theta'$ was not available within the dataset utilized (which had already been 5-minute averaged) and is therefore ignored for this study. Previous analysis, however, has shown that $\theta'$ varies inversely with $U$ in complex terrain (Papadopoulos et al., 1992), and we may therefore assume its influence is largely captured by $U$.

Although these unadulterated features give us an idea as to how the system is working at the moment, they may not explicitly represent the relevant atmospheric forcing mechanisms. Our list of measurements allows us to break down our system into two principal forcing components: buoyancy and inertial forcing (which indirectly includes pressure gradient forces). Each of these

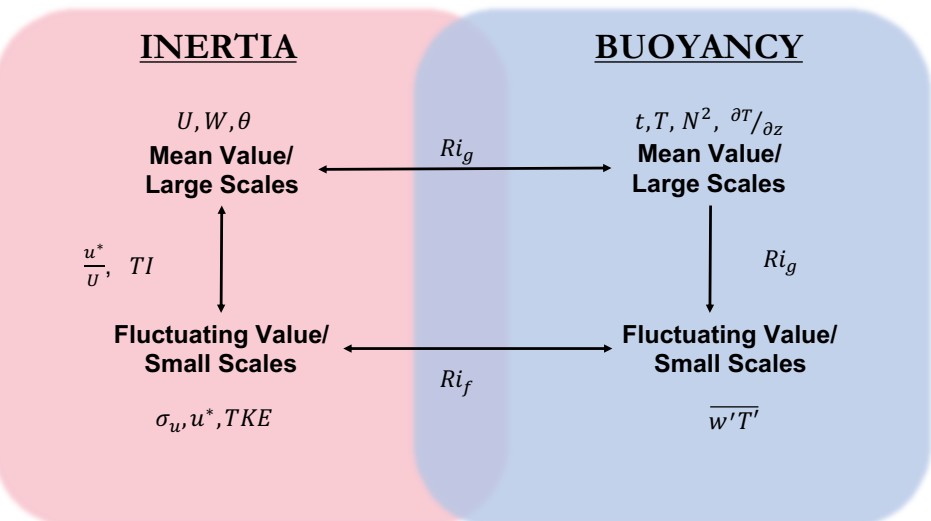

**Figure 2.** Illustrative breakdown of the scales and variables related to inertial and buoyant forcing. $\theta'$ is not shown as it is not utilized in the analysis.

forces can be further discretized into large and small scales (also called mean and fluctuating values; typically separated by at least one order of magnitude).

Fig. 2 shows an illustrative breakdown of the two main forcing mechanisms alongside a list of extracted descriptor variables. The definitions and formulations of all non-obvious extracted variables used in this study can be found in Appendix A. From
175 this figure, it is clear that the variables in Eqn. 3, when manipulated, are able to describe both the inertial and buoyant forces at multiple scales. Large-scale inertial forcing can be described by the local mean wind speed and direction ($U$ and $\theta$) or vertical velocity $W$, while small-scale inertial forcing can be described by variables such as the fluctuating (standard deviation of) velocity $\sigma_u$, friction velocity $u^*$, and the turbulence kinetic energy $TKE$. Likewise, large-scale buoyancy forcing can be described by the squared buoyancy frequency $N^2$, the temperature gradient $\partial T / \partial z$, or by proxy values such as the time of
180 day $t$ or temperature $T$ (which, on average, is higher during the day and lower at night; stability parameters based on Monin-Obukhov similarity theory have been considered ill-suited for complex terrain flows because of the breakdown of underlying assumptions (Fernando et al., 2015), and hence were not used in this study). Small-scale buoyancy effects can be described by the turbulent heat flux $\overline{w'T'}$. The correspondence between forces and internal parameters can also be described by non-dimensional variables such as the gradient Richardson number $Ri_g$, flux Richardson number $Ri_f$, turbulence intensity $TI$, and
185 normalized friction velocity $u^*/U$. These derived non-dimensional variables, or extracted features, are typically ignored by current ML models in lieu of raw features such as those listed in Eqn. 3.

Extracted variables like those in Fig. 2 may not provide any more information than the raw variables in Eqn. 3. However, they may ease the burden on the model by discretizing (or directly relating) informational categories, therefore reducing informational overlap and noise, providing more periodic/predictive power, and more accurately describing the underlying system.

Further, such well-conceived meteorological variables have been seen to be useful for atmospheric prediction (Kronebach, 1964; Li et al., 2019). In theory, given enough data, the model should be able to decipher and interpret these extracted features on its own. Unfortunately there often isn't enough collected data for this to happen organically. Instead, by providing better information we can create a simpler, cheaper, more robust model that requires less training data and construction time. Selected features will ideally represent the underlying system as accurately as possible without providing noisy or redundant information.

## 3.4 Testing

In an effort to understand the predictive capabilities of each variable, initial tests only include individual atmospheric input features. Once each input feature has been tested separately, a feature set that utilizes all input features is tested. Feature importance estimates are then extracted from the random forest model and various user-selected combinations of the most important input features are tested. It must be noted that only select input feature sets were tested in this investigation due to the sheer multitude of potential feature sets.

In order to relieve any timescale bias, forecasts are made across multiple timescales. Typically, wind power utility operators require single-step short range power forecasts run hour-by-hour for a few days to reduce unit commitment costs. The forecast skill of observation-based methods generally reduces with forecast lead time within an hour, and numerical models have higher skill in forecasting larger time leads (> 3 hours) (Haupt et al., 2014). Statistical learning methods have proved to be particularly effective from about 30 minutes to approximately three hours ahead (Mellit, 2008; Wang et al., 2012; Yang et al., 2012; Morf, 2014), and roughly this time frame is thus the focus for this study. The shortest forecast predicts wind speeds 10 minutes ahead, roughly within the turbulent spectral band (Van der Hoven, 1957). Forecasts are also made one and three hours ahead, which are within the spectral gap between the turbulent and synoptic spectra and approach the six-hour period wherein NWP models become particularly useful (Dupré et al., 2019). These are all single-step forecasts, which is to say that the averaging timescale increases with the forecasting timescale (e.g. a 10-minute forecast predicts 10-minute averaged wind speed, whereas a three-hour forecast predicts three-hour averaged wind speed). Each test is performed 10 times to ensure forecasting stability.

Two metrics are utilized to determine how well the random forest predicts exogenous error. The root mean squared error (RMSE) of the bias-corrected ARIMA model is found, giving a metric of the true exogenous error. The random forest model is then trained to predict the exogenous error, combined with the ARIMA model, and the newly constructed ARIMA-RF is used to forecast wind speeds. The reduction in RMSE (which comes exclusively from the random forest's prediction of exogenous error) is then found for the test set. The coefficient of determination ($R^2$) between the true and predicted exogenous error is used to determine the amount of error variability captured by the random forest model. Eqn. 4 and Eqn. 5 describe both metrics,

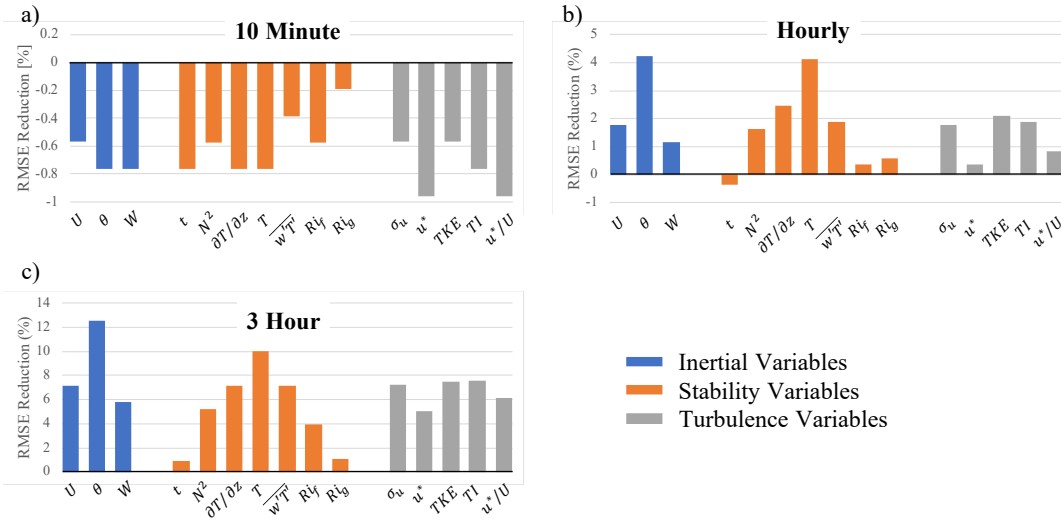

**Figure 3.** Percent reduction (or increase) in RMSE obtained by the random forest model when given select meteorological inputs. Blue, orange, and grey bars represent inertial, stability, and turbulence input features, respectively.

wherein $U_m$ is the target wind speed, $\hat{U}_m$ the predicted wind speed, $\varepsilon'_m$ the true exogenous error, $\hat{\varepsilon}'_m$ the predicted exogenous error, $\overline{\varepsilon'}$ the mean exogenous error (approximately zero), $m$ each individual sample, and $M$ the sample size.

$$RMSE = \sqrt{\frac{1}{M}\sum_{m=1}^{M}(U_m - \hat{U}_m)^2} \tag{4}$$

$$R^2 = 1 - \frac{\sum_{m=1}^{M}(\varepsilon'_m - \hat{\varepsilon}'_m)^2}{\sum_{m=1}^{M}(\varepsilon'_m - \overline{\varepsilon'})^2} \tag{5}$$

## 4  Results

Fig. 3 shows the reduction (or increase) in forecasting RMSE obtained via the random forest model for each individual input feature. Specific RMSE and $R^2$ values obtained for these cases may be found in Table B1 in Appendix B. The variables are broken down into three distinct categories: inertial (large scale dimensional variables signifying inertial forces in Fig. 2), stability (blue and purple regions in Fig. 2 which are akin to atmospheric stability), and turbulence variables (small scale and non-dimensional inertial variables in Fig. 2). It is immediately clear that there is a distinction between the results for the 10-minute forecast and those for the hourly and three-hour forecasts. Each random forest prediction of 10-minute exogenous error using individual input features resulted in an increase in RMSE (or negative RMSE reduction; Fig. 3a), indicating that

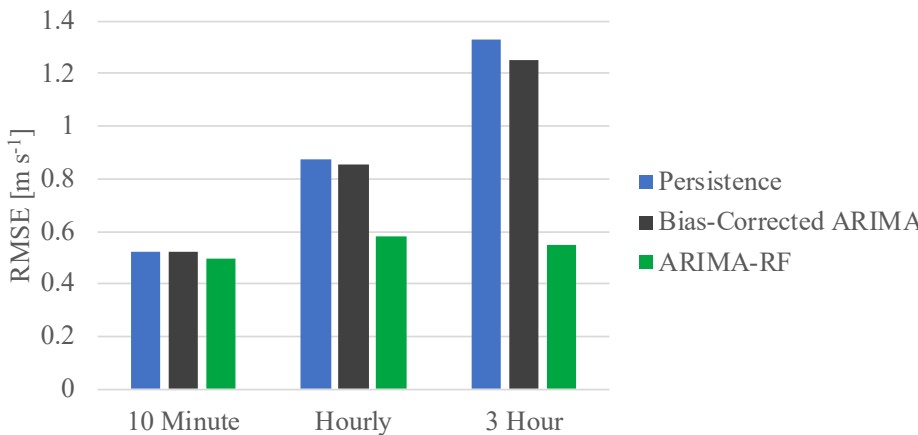

**Figure 4.** Comparison of RMSE obtained by the persistence, bias-corrected ARIMA, and ARIMA-RF with all meteorological inputs for all forecasting timescales.

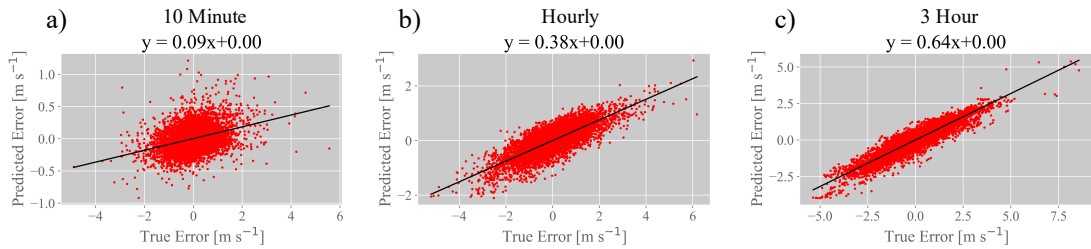

**Figure 5.** Correlation between true and predicted exogenous error using all input features. a) shows correlation for the 10-minute prediction, b) for the hourly prediction, and c) for the three-hour prediction. Black line denotes the best-fit line, an equation for which is given above each plot. Corresponding $R^2$ values are given in the bottom row of Table B2.

exogenous error at such small timescales is highly chaotic and unpredictable based off of the information from any single atmospheric variable. In fact, these tests show that any correlative patterns observed between the utilized meteorological variables and exogenous error are likely circumstantial and lead to deleterious predictions.

Fig. 3b and c show reduction in RMSE for hourly and three-hour forecasts, respectively. Both $\theta$ and $T$ appear to be the most beneficial individual input features at these timescales, while $t$ and $Ri_g$ are the least helpful. $TI$, $\sigma_u$, and $TKE$ are the most beneficial turbulence variables and provide similar levels of improvement at both the hourly and three-hour timescales. Interestingly, turbulence variables as a group continue to provide valuable information even for multi-hour forecasting timesteps. The heterogeneity of improvement (over all individual input features) increases with prediction timescale, with $\theta$ reducing exogenous error by over 12% for the three-hour forecast.

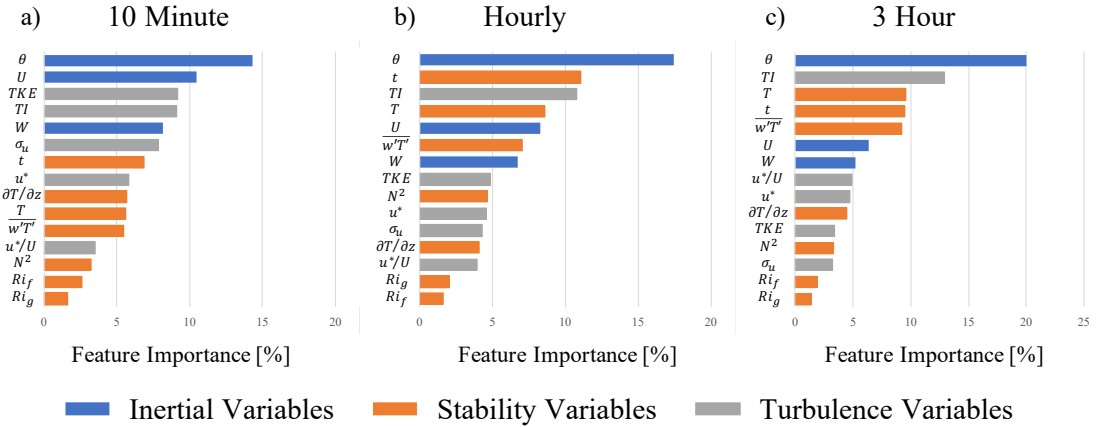

**Figure 6.** Feature importance for the prediction of exogenous error when all input features are given to the random forest model. a) shows importance for the 10-minute prediction, b) for the hourly prediction, and c) for the three-hour prediction. Blue bars denote inertial variables, orange denote stability variables, and grey bars denote turbulence variables. Importance values for each test sum to 100%.

Utilizing all input features within the random forest resulted in drastic improvements in exogenous error prediction. Fig. 4 shows a comparison of the RMSE obtained by the ARIMA-RF model to that obtained by the persistence and bias-corrected ARIMA models. The bias-corrected ARIMA model's RMSE amounted to 0.523, 0.852, and 1.251 m s$^{-1}$ for the 10-minute, hourly, and three-hour forecasts, respectively. The random forest model, utilizing all input features, reduced these RMSE values by 7%, 32%, and 56%, respectively (RMSE values given in Table B2 in Appendix B). The correlation between true and predicted exogenous error can be seen in Fig. 5. It is clear that, as prediction timescale increases, the correlation between true and predicted exogenous error increases, with the three-hour prediction having an R$^2$ value of 0.801.

Feature importance estimates were also obtained from the all-input test cases and can be seen in Fig. 6. A handful of variables, namely $\theta$, $U$, $TI$, $t$, $T$, and $\overline{w'T'}$, are particularly useful for the hourly and three-hour predictions. Because $U$, $\theta$, and $t$ are all variables that can be obtained from a simple cup anemometer and wind vane, they are used as the "base variables" when testing discriminate input feature combinations. The results of these tests, which may be found in Table B2 in Appendix B, prove that a large majority of the model's predictive power (i.e. a majority of the relevant input information) is contained within these six variables.

## 5 Discussion

There is a clear distinction between the results obtained for the 10-minute exogenous error predictions and those obtained for the hourly and three-hour predictions. All atmospheric input features, when used individually for the 10-minute forecasts, resulted in a faulty prediction of error. This is likely due to the highly chaotic nature of wind speeds at the 10-minute timescale. Typically the large-eddy turnover timescale for the lower atmosphere is 10-20 minutes (specifically during daytime), and

averaging timescales approaching or less than this timescale exclude information on more stable and deterministic large eddies, thus making predictions more prone to random errors. This is exemplified by the work of Van der Hoven (1957), who shows that a 10-minute average is within the turbulent peak of the wind speed spectrum. The lack of large eddy influence results in a wind speed signal that is replete with random fluctuations originating in the inertial subrange, adding substantial noise to the prediction. These fluctuations overwhelm the ML model's pattern recognition capabilities, reducing the random forest prediction to a noisy guess. Such ML models will always make predictions based on patterns in the training data, even when those patterns are erroneous and do not hold for the testing dataset. This results in error predictions that are not correlated with the true exogenous error (as indicated by 10-minute $R^2$ values in Table B1).

As the forecasting timescales increase, smaller-scale turbulent fluctuations average out and the random forest model can recognize predictive patterns between atmospheric input features and the non-linear exogenous error. Tests involving individual atmospheric variables effectively represent the magnitude of the first term on the right side of Eqn. 1. These tests show that predictions involving individual variables (or at least those tested) can only reduce exogenous error by approximately 4% and 12% for the hourly and three-hour predictions, respectively. While this is a considerable error reduction, the meteorological variables are most beneficial when utilized in unison.

A list of feature importance estimates, as determined by a test incorporating all input features, is shown in Fig. 6. Many of the features are correlated, meaning that exact importance values are likely misleading. Nevertheless, the reported importance estimates are likely a good indicator as to which features, when used in combination with others, are most useful in predicting exogenous error. $\theta$ is both the best individual predictor and the most important feature for all tests, likely because our measurements are taken atop an asymmetric ridge in complex terrain. As is detailed in Fernando et al. (2019), the complex terrain leads to an ensemble of topographically induced ridge-top flow features such as jetting, mountain waves, and reversed flows which have a large impact at the measurement location.

Fig. 7 shows how the ARIMA-RF model (utilizing the full input feature set) performs across the domain of an integral set of inertial, turbulence, and stability input features. The 10-minute prediction performs best at wind speeds up to 4 m s$^{-1}$ (Fig. 7a). Above this limit, the model's RMSE gradually increases with increasing wind speed. Hourly and (particularly) three-hour predictions perform worse than the 10-minute predictions for wind speeds below 3 m s$^{-1}$. However, both models are most accurate at moderate wind speeds between 3-7 m s$^{-1}$. Faster wind speeds ($\geq$8 m s$^{-1}$) tend to cause an increase in RMSE for all three models, perhaps due to a relatively low sample size. Wind speeds between 3-7 m s$^{-1}$ make up more than 50% of the observed periods, whereas wind speeds $\geq$ 8 m s$^{-1}$ make up less than 20% of the periods. All models observed are accurate to within 0.7 m s$^{-1}$ in the operating region of most wind turbines (4 – 12 m s$^{-1}$; RMSE values above this limit are not shown due to a statistically insignificant number of testing samples). The ARIMA-RF hourly forecast obtains a correlation coefficient of 0.71 with the true wind speed, akin to that of numerical models in complex terrain (Yang et al., 2013).

The ARIMA-RF model's accuracy as a function of time is shown in Fig. 7b. The difference between the 10-minute and hourly/three-hour forecasts is apparent, as the former is more accurate during nocturnal conditions because of the smaller integral timescale of turbulence ($\sim O(1)$ minute) whereas the latter is most accurate during the afternoon hours (integral timescales $\sim O(10)$ minutes). This discrepancy is largely based upon atmospheric stability, as the 10-minute prediction is

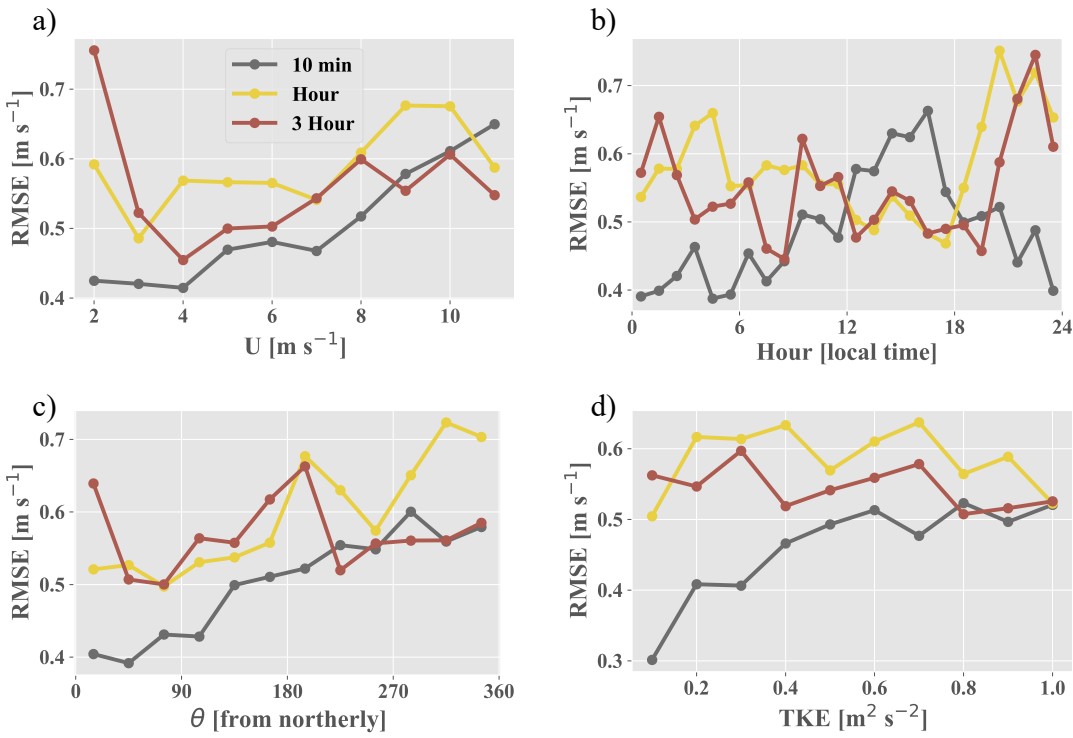

**Figure 7.** RMSE obtained by the ARIMA-RF tests incorporating all input features partitioned by (a) wind speed, (b) hour of the day (local time), (c) wind direction, and (d) $TKE$.

~10% more accurate during stable periods than unstable; the opposite is true for hourly and three-hour timescales, which perform 17% and 9% better, respectively, during stable periods (Table B3 in Appendix B). Relatively high turbulence during the daytime clearly hampers the model when forecasting 10 minutes ahead (Fig. 7d). However, as these fluctuations average out over larger timescales, the model is able to more accurately predict future wind speeds. Interestingly, the model struggles to predict an hour or more ahead during stable conditions, as the RMSE of both the hourly and three-hour models spike during the nocturnal transition (sunset typically between 2000 – 2100 local time). This spike in RMSE coincides with peak wind ramp hours (defined as wind speed changes of 20% and 50% for hourly and three-hour forecasts, respectively) which tend to occur between 1900 – 2300 local time (not shown). Stable atmospheric conditions can lead to phenomena such as mountain waves and flow jetting (Fernando et al., 2019; Vassallo et al., 2020b), features which could lead to such wind ramp events and would be difficult for the statistical models to predict 1-3 hours ahead.

Fig. 7c shows that the ARIMA-RF model performs significantly better for northeasterly flows compared to westerly flows. This discrepancy, particularly on the 10-minute timescale, is a result of complex topography upstream (during periods with westerly winds) which tends to create turbulent bursts that are averaged out at larger lead times. Fig. 7d shows that 10-minute forecasts perform approximately 40% better during low $TKE$ ($\leq 0.1$ m$^2$ s$^{-2}$) periods compared to high $TKE$ ($\geq 0.8$ m$^2$

$s^{-2}$) periods. However, the nearly constant RMSE for hourly and three-hour forecasts shows that the ARMIA-RF model is not affected by varying stochastic processes at larger averaging scales. There is a clear point of directional discontinuity on the 10-minute timescale, as the model performs drastically better when wind is north-northeasterly (NNE; 0-30°) as opposed to north-northwesterly (NNW; 330-360°). This can be explained by the fact that NNW flows tend to exhibit much higher TKE values than do NNE flows (Fig. B1 in Appendix B). Many of the input features are clearly interrelated, adding another layer of complexity to the prediction process and further emphasizing the need to extract necessary meteorological information via prudent feature engineering.

The six most important features for the hourly and three-hour predictions are identical (although scrambled), and were therefore used to test discriminate feature set combinations. All tests with multiple input features contained $U$, $\theta$, and $t$. There are two reasons for prioritizing these three variables: they prove to be some of the most important input features for all timescales (Fig. 6) and they can all be captured by a simple cup anemometer and wind vane rather than a more expensive sonic anemometer. These three features, when used in conjunction, were able to capture about 66% of the maximum error reduction seen for all timescales. Discriminate input sets incorporating only $U$, $\theta$, $t$, $TI$, $\overline{w'T'}$, and $T$ are able to capture over 90% of the exogenous error caught by the tests incorporating all input features, indicating that almost all of the relevant information in our inputs can be retrieved from these six variables. Notably, many of the most important input features ($U$, $\theta$, $t$, $T$, and $W$) are directly measurable and need not be extracted (although $T$ and $W$ cannot be captured by a cup anemometer). The most important variables that require extraction (i.e. values that are not direct measurements), $TI$, $TKE$, and $\overline{w'T'}$, all contain small-scale (fluctuating) forcing components, indicating that small-scale processes may be more easily captured by ML models after domain-specific interpretation. These small-scale variables provide significant predictive power, even at a multi-hour timescale. The testing results from the study show that, in order to achieve an optimal forecast of exogenous error, these small scales must be included as an input for the predictive model.

Tests combining multiple atmospheric variables are particularly useful because they incorporate the second term on the right side of Eqn. 1, an indication of how the exogenous error changes depending on the input features' co-variance. This is especially true for the testing case incorporating all input features. As expected, this case provided the best predictions of exogenous error. The correlation between the predicted and true exogenous error (Fig. 5) dramatically increases with increasing timescales, with the best three-hour random forest prediction capturing 80% of exogenous error variability. As Fig. 4 shows, the best ARIMA-RF error is roughly 0.5 m s$^{-1}$ for all timescales even though both the persistence and bias-corrected ARIMA models get worse as timescales increase. This is an encouraging result, in that meteorological forecasting models need not necessarily get worse with time (although the averaging timescales likely must increase proportionately). Exogenous error prediction gets far better with increasing timescales, with the best random forest prediction reducing forecasting RMSE by over 50%. There appears to be a floor (0.5 m s$^{-1}$) on the predictability of exogenous error, indicating that there may be certain atmospheric information missing from the set of input features. This information could come from other external forces or could be a result of forcing at scales that have not been captured by our current input feature set.

## 6 Conclusions

Exogenous error arises from atmospheric forcing that is ignored or misrepresented in the modeling process. It has been shown that this error, or a portion thereof, can be predicted by an ML model given relevant atmospheric information. $\theta$ and $T$ were found to be particularly beneficial as individual inputs, while the combination of $U$, $\theta$, and $t$, features which may be derived from a simple cup anemometer and weather vane, were able to provide a majority of the maximum error reduction seen at every timescale. Domain-specific feature extraction was found to be particularly useful for input features relating small-scale forcing, and these turbulence variables were found to have significant predictive power even for multi-hour forecasts. The lowest RMSE value was relatively constant at all prediction timescales, indicating that there is additional relevant atmospheric information that this list of inputs does not capture. The results are promising, however, in that they illustrate that forecasting accuracy need not decrease at large timescales. In fact, at large timescales turbulent fluctuations average out, allowing mesoscale and synoptic forces to provide a clearer signal for exogenous error prediction.

While the exact results of this investigation are site-specific, the findings are expected to be generally applicable to numerous wind projects, especially those located in complex terrain. Prudent implementation of atmospheric forcing information, particularly that which is non-linear or derived via coupling of multiple forces, is crucial for the prediction of exogenous error and must be addressed to obtain optimal forecasting results. This study supports the supposition that a hybrid model using ML techniques to correct a simpler statistical predictor (such as an ARIMA model) can be effective for wind speed forecasting.

Further improvements are still required to more accurately represent atmospheric forcing. Gridded meso or synoptic-scale information would allow the model to predict transitional periods including weather fronts and drastic wind ramp events. Multiple scales of forcing should also be incorporated to improve the pattern recognition capabilities of ML techniques. Additional information about microscale, mesoscale, and synoptic events would better depict atmospheric forcing and momentum, and the effects of seasonality must be accounted for when possible. It is also worth exploring the model's capabilities when the dataset is not randomly shuffled (i.e. whether a model trained on past years' data can accurately predict exogenous error over an entire year). Hopefully, this study will be a forerunner for the improved incorporation of atmospheric physics within ML modeling.

*Code and data availability.* Data from the Perdigão campaign may be found at https://perdigao.fe.up.pt/. Due to the multiplicity of cases analyzed in this study, example processing and modeling codes can be found at https://github.com/dvassall/.

## Appendix A: Input Features

Atmospheric variables were measured using sonic anemometers and temperature sensors along a single 100 m tower. When possible, missing data from the 100 m sensors were filled via correlation with the 20 m sensors using the variance ratio measure-correlate-predict method (Rogers et al., 2005). There were no periods with functional 100 m sensors and nonfunctional 20 m sensors. All periods without any measurements from both sets of sensors (15 5-minute periods) were filled using linear

regression with Gaussian white noise. Many of the input features used in the study required derivation. A description of necessary derivations are given below.

Friction velocity is defined as $u^* = \left(\overline{u'w'}^2 + \overline{v'w'}^2\right)^{1/4}$ and was measured at 20 m AGL, just above canopy height (Fernando et al., 2019). Turbulence kinetic energy is defined as $TKE = \frac{\overline{u'^2} + \overline{v'^2} + \overline{w'^2}}{2}$ and was measured at 100 m AGL. Buoyancy frequency squared is typically defined as (see Kaimal and Finnigan (1994) for details of all parameters that appear below)

$$N^2 = \frac{g}{\rho_0}\frac{\partial \rho}{\partial z} = \frac{g}{T_{pv0}}\frac{\partial T_{pv}}{\partial z} \tag{A1}$$

where $g$ is the gravitational force, $\rho$ the air density, $z$ the height AGL, $T_{pv}$ the virtual potential temperature, and subscript 0 indicates reference variables in using the Boussinesq approximation. The gradient Richardson number is defined as

$$Ri_g = \frac{N^2}{\left(\frac{\partial u}{\partial z}\right)^2 + \left(\frac{\partial v}{\partial z}\right)^2} \tag{A2}$$

where $u$ and $v$ are the two horizontal wind speed components. The flux Richardson number is defined as

$$Ri_f = \frac{\frac{g}{T_v}\overline{w'T'}}{\overline{u'w'}\left(\frac{\partial u}{\partial z}\right) + \overline{v'w'}\left(\frac{\partial v}{\partial z}\right)}, \tag{A3}$$

where $T_v$ is the virtual temperature while $\overline{u'w'}$ and $\overline{v'w'}$ (both measured at 100 m AGL alongside $\overline{w'T'}$ and $T_v$) are the Reynolds stresses that indicate the flow's vertical momentum flux. $Ri_f$ is typically used in conjunction with a stably stratified atmosphere (Lozovatsky and Fernando, 2013). It is used here in the general sense as it is a measure of the ratio between buoyant energy production and mechanical energy production (associated with inertial forces) related to Fig. 2. Negative $N^2$ values, corresponding to convective atmospheric conditions, are set to 0. $Ri_g$ and $Ri_f$ are limited to a maximum of 5 and minimum values of 0 and $-5$, respectively, to remove extremes in both variables. Turbulence intensity is the ratio of fluctuating to mean wind speed, or $TI = \sigma_u/U$. Both hour of the day and wind speed were broken into two oscillating components in order to eliminate any temporal or directional discontinuity.

## Appendix B: Testing Results & Analysis

Table B1 presents the RMSE obtained by the bias-corrected ARIMA model (total exogenous error) and the ARIMA-RF using individual features. $R^2$ values denote the correlation between the true and predicted exogenous error. Features are separated into inertial (top), stability (middle), and turbulence (bottom) inputs as described in Section 4. Table B2 presents the RMSE values obtained by the persistence and bias-corrected ARIMA models alongside the RMSE and $R^2$ (between true and predicted exogenous error) values obtained by the ARIMA-RF while utilizing input feature combinations that are of particular interest. The final row in Table B2 shows the results of the ARIMA-RF when all input features are utilized.

Table B3 shows how the ARIMA-RF model (with the full input feature set) performs based on atmospheric stability. Stable periods are defined as those which have $N^2$ values greater than zero, unstable as those which have $N^2$ values of zero.

Fig. B1 shows average $TKE$ partitioned by direction (30° bins) for 10-minute periods. Northeasterly flows display the lowest $TKE$ values, whereas westerly flows display the highest average $TKE$.

| Model/Input | 10 Minute | | Hourly | | 3 Hour | |
|---|---|---|---|---|---|---|
| | RMSE | $R^2$ | RMSE | $R^2$ | RMSE | $R^2$ |
| Bias-corrected ARIMA | 0.523 | - | 0.852 | - | 1.251 | - |
| $U$ | 0.526 | -0.005 | 0.837 | 0.033 | 1.162 | 0.129 |
| $\theta$ | 0.527 | -0.004 | 0.816 | 0.075 | 1.094 | 0.220 |
| $W$ | 0.527 | -0.007 | 0.842 | 0.022 | 1.179 | 0.093 |
| $t$ | 0.527 | -0.013 | 0.855 | 0.003 | 1.240 | 0.021 |
| $N^2$ | 0.526 | -0.008 | 0.838 | 0.034 | 1.186 | 0.097 |
| $\partial T/\partial z$ | 0.527 | -0.012 | 0.831 | 0.040 | 1.162 | 0.129 |
| $T$ | 0.527 | -0.005 | 0.817 | 0.078 | 1.126 | 0.174 |
| $\overline{w'T'}$ | 0.525 | -0.006 | 0.836 | 0.035 | 1.162 | 0.137 |
| $Ri_f$ | 0.526 | -0.008 | 0.849 | 0.012 | 1.202 | 0.082 |
| $Ri_g$ | 0.524 | 0 | 0.847 | 0.010 | 1.238 | 0.025 |
| $\sigma_u$ | 0.526 | -0.016 | 0.837 | 0.027 | 1.160 | 0.143 |
| $u^*$ | 0.528 | -0.017 | 0.849 | 0.014 | 1.188 | 0.081 |
| $TKE$ | 0.526 | -0.014 | 0.834 | 0.039 | 1.157 | 0.149 |
| $TI$ | 0.527 | -0.008 | 0.836 | 0.038 | 1.156 | 0.160 |
| $u^*/U$ | 0.528 | -0.008 | 0.845 | 0.023 | 1.174 | 0.109 |

**Table B1.** The top row shows RMSE (m s$^{-1}$) obtained by the bias-corrected ARIMA model. Below are the resulting RMSE and $R^2$ (between true and predicted exogenous error) values from ARIMA-RF predictions utilizing individual inputs for all forecasting timescales. Input features are separated into inertial (top), stability (middle), and turbulence (bottom) variables, as described in Section 4.

| | 10 Minute | | Hourly | | 3 Hour | |
|---|---|---|---|---|---|---|
| **Model** | **RMSE** | **$R^2$** | **RMSE** | **$R^2$** | **RMSE** | **$R^2$** |
| Persistence | 0.525 | - | 0.873 | - | 1.326 | - |
| Bias-corrected ARIMA | 0.523 | - | 0.852 | - | 1.251 | - |
| **Input Features** | **RMSE** | **$R^2$** | **RMSE** | **$R^2$** | **RMSE** | **$R^2$** |
| $U, \theta, t$ | 0.501 | 0.076 | 0.672 | 0.369 | 0.750 | 0.618 |
| $U, \theta, t, T$ | 0.496 | 0.096 | 0.628 | 0.453 | 0.657 | 0.711 |
| $U, \theta, t, TI$ | 0.495 | 0.099 | 0.643 | 0.424 | 0.694 | 0.681 |
| $U, \theta, t, \overline{w'T'}$ | 0.497 | 0.087 | 0.651 | 0.404 | 0.704 | 0.665 |
| $U, \theta, t, TI, \overline{w'T'}, T$ | 0.490 | 0.116 | 0.606 | 0.491 | 0.610 | 0.755 |
| All input features | 0.489 | 0.116 | 0.581 | 0.533 | 0.549 | 0.801 |

**Table B2.** RMSE (m s$^{-1}$) obtained by the persistence and bias-corrected ARIMA models as well as the RMSE obtained by the ARIMA-RF when utilizing select input feature combinations. $R^2$ values between true (defined as the bias-corrected ARIMA error) and predicted exogenous error is also reported for each test case. The final row shows the final test which uses all input features.

| **Timescale** | **Stable** | **Unstable** |
|---|---|---|
| 10 minutes | 0.530 | 0.476 |
| 1 Hour | 0.504 | 0.606 |
| 3 Hours | 0.511 | 0.561 |

**Table B3.** RMSE (m s$^{-1}$) obtained by the ARIMA-RF model (with the full input feature set) based on stability (as defined by $N^2$) of the forecasted time period.

*Author contributions.* Daniel Vassallo prepared the manuscript with the help of all co-authors. Data processing was performed by Daniel Vassallo, with technical assistance from Raghavendra Krishnamurthy. All authors worked equally in the manuscript review process.

*Competing interests.* The authors declare that they have no conflict of interest.

*Acknowledgements.* This work was funded by the National Science Grant numbers AGS-1565535 and AGS-1921554, Wayne and Diana Murdy Endowment at University of Notre Dame and Dean's Graduate Fellowship for Daniel Vassallo. The Pacific Northwest National Laboratory is operated for the DOE by Battelle Memorial Institute under Contract DE-AC05-76RLO1830. Special thanks to the teams at both EOL/NCAR and DTU who collected and managed the tower data utilized in this study.

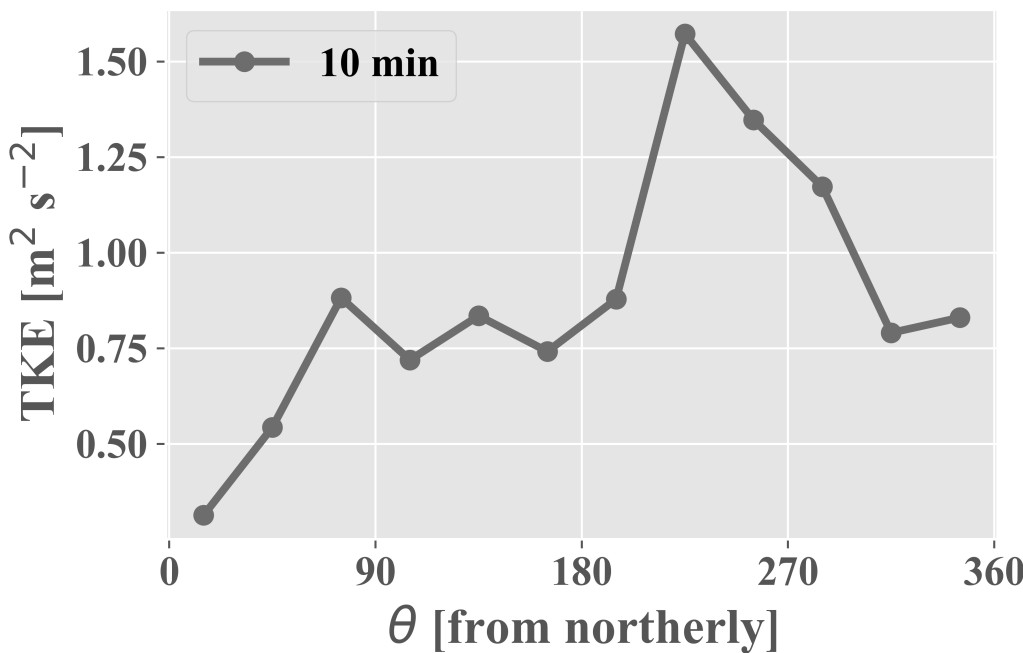

**Figure B1.** Average 10-minute TKE by incoming flow direction. Wind direction is partitioned into $30°$ bins.

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
