# Peer review of "Utilizing Physics-Based Input Features within a Machine Learning Model to Predict Wind Speed Forecasting Error"

_Wind Energy Science, 2020_

## Referee Comment (RC1) · Javier Sanz Rodrigo (Referee) · 25 May 2020

Very interesting work investigating the potential use of atmospheric quantities measured at a met mast to improve very-short-term predictions of wind speed from 10 min to 3 hr ahead using machine learning. The relevance of different input feutures in isolation and combined is quantified showing significant error reductions with quantities that can be easily extracted from conventional mast measurement campaigns making the study particularly useful for implementation in operational tools. The testing methodology is convincing only requiring some clarifications. In particular, the selection of the input averaging interval of 5 min requires additional motivation. Have the authors

tested different input intervals to see the impact in the error reduction?

Comments:

Page 4, line 7: Can you elaborate further on how wind speed data is stationary? Is this tested at the prdiction timescales (10 min - 3hr)? I would also expect wind speed to be subject to seasonal and diurnal variability. Please clarify.

P5, L28: I'm curious why are you not using the Obukhov length (or z/L)? Isn't it a more commonly used parameter to characterize stability? You may want to motivate this selection even though from the results of Figure 6 it seems that stability parameters are not that important in the improvement of forecasts.

P7, L28: How are sonic measurements corrected for tilt? What is the interval used when deriving the fluxes? Is it equal, shorter or longer than the 5-min interval used in the moving average? This is just to know if 5-min is the actual filter in the data or if the data already came with a longer averaging time. This could also be relevant to understand the potential impact of this filter in the performance at 10 min prediction horizon (Figure 3).

P8, Figure 2: The map is difficult to read. It would be better to show a elevation contour plot where we can read the relative heights. I don't think it is necessary to provide an illustration of the mast levels if they are described in the text.

P8, L8: You end up using 5-min averaged data to build predictive models with prediction horizonts at 10 min, 1 hour and 3 hours. You previously mentioned that these are single-step forecasts. Wouldn't you have to use input data that is averaged at the same interval than the forecast step (e.g. use 3-hour moving averages to predict 3 hr ahead)? Or do you forecast {10min, 1hr, 3h} ahead based always based on 5-min data? If the latter is true, please clarify why not using a consistent interval between input and prediction data or, alternatively, how dependent are the results to the chosen interval in the time series.

[Figure]

P10. Figure 4: One may wonder how a Persistance-RF model would work. This might be a good result to include in the paper so that you can just isolate the impact of RF from that of the forecasting model to make the results more generally applicable. Maybe you get to the same conclusions with a simpler model.

P8. L11: How is the flux Richardson calculated between 100-20 m? Isn't it a local quantity derived from a sonic level? Is it the mean value between the two levels? Please clarify.

---

## Referee Comment (RC2) · Anonymous Referee #2 · 6 Jun 2020

**Utilizing Physics-Based Input Features within a Machine Learning Model to Predict Wind Speed Forecasting Error**

*Daniel Vassallo, Raghavendra Krishnamurthy and Harindra J.S. Fernando*

**REVIEW**

**GENERAL COMMENT:**

The paper by Vassallo et al. presents an interesting contribution that uses a ML approach to predict wind speed forecasting error. The paper is overall well-written, and the methods used described in detail. The final discussion of the results is also well-laid. Nevertheless, I have some major points that require clarification before the paper can be recommended for publication. In addition, the analysis presented in the current version of the manuscript, though interesting, could definitely be expanded given the unique dataset the authors are considering.

**MAJOR COMMENTS:**

1.  I find the choice of the authors of (extensively) describing the methods used before the data a bit confusing. I personally had to go back to the methods section after reading the data section to make sure I got everything right. I would recommend switching the order of the two sections.
2.  More clarification is needed on what averaging time is used in the calculations of the variables considered in this work. For example, what averaging time is used to calculate the Reynolds decomposition for turbulent averaging, for example for TKE, TI, friction velocity? Why did you choose it? How does that conciliate with the different lead times of the ML models?
3.  In addition, the authors should better clarify how the random split of train-test set mentioned in the paper is implemented. Do you mean that you are randomly picking 25% of data for testing, and then using those times stamps, once the algorithm has been trained, to predict wind speed 10-min, 1-hour, 3-hour ahead of each of the randomly selected time stamps in the testing?
4.  With such a huge data set as the one used in this analysis, I feel like the results shown could be greatly expanded, as a lot of additional analysis relevant to the topic could be made. After all, you are using 2 sonic anemometers out of an array of almost 200. For example, how does the performance of the used ML algorithms vary with atmospheric stability? Or with height? Do you find that different input features are more relevant close to the surface compared that at let's say hub-height? Or how do the results vary in different complex terrain locations, for example comparing results from the valley and from the ridge tops? Please consider adding more analysis to this piece of work.

**SPECIFIC COMMENTS:**

5. Abstract: introducing the symbols of each feature are not necessary in the abstract.
6. Figure 2-a: the map is not super clear.
7. Figure 2-b: not really needed.
8. Page 4: was wind speed at Perdigão really stationary? Over which time scales? Please clarify.
9. P.6 l.10: rephrase as "a feature set that utilizes all input features is tested".
10. Did you apply any cross-validation for your ML models? If not, why?
11. P.8 l.1: please specify what you mean by "Sensors at 20 and 100 m AGL were chosen based on data availability."
12. Please state the native time resolution of the sonic data you are using.
13. Have sonic anemometer data been filtered for tower wake effects? These effects would artificially increase turbulence (and reduce wind speed) for some wind direction bins, thus invalidating the quality of quite some data.

---

## Author Comment (AC1) · 30 Jul 2020

We would like to thank both referees for their help in improving the quality of the manuscript. Please find attached the authors' response to the referees' comments as well as a copy of the manuscript with the changes highlighted.

Please also note the supplement to this comment:
https://wes.copernicus.org/preprints/wes-2020-61/wes-2020-61-AC1-supplement.zip
* * *

---

## Author Response (AR1)

Ref: wes-2019-58
Title: Utilizing Physics-Based Input Features within a Machine Learning Model to Predict Wind Speed Forecasting Error
Journal: Wind Energy Science

**Referee:** Javier Sanz Rodrigo

We would like to thank Dr. Rodrigo for his time in reading and commenting on the manuscript that led to considerable improvement of the paper. We have tried to address all comments and hope that this revision is acceptable for publication.

**Have the authors tested different input intervals to see the impact in the error reduction?**

The utilized data had already been preprocessed and 5-minute averaged by NCAR; we had forgotten to include this information in the original and have added a reference on P3 L28.

5-minute averaging is a common averaging period used in most meteorological studies (for example CASES-99, RASEX, Perdigão, etc.) as it helps minimize flux sampling errors (systematic, random, and mesoscale variability error) and provides necessary flags to categorically distinguish between instrumental problems and plausible physical behavior (Mahrt et al. 1996, Sun et al. 1996, Vickers and Mahrt 1997). A local average of 5 minutes seems to adequately capture most of the turbulent fluxes in stationary time periods compared to one-hour local averaging (Mahrt et al., 1996, Sun et al., 1996). A 20 to 30-minute time-averaging protocol has become standard eddy-covariance practice for idealized conditions (i.e., quasi-stationary and horizontally homogeneous), but one can combine these 5-minute averages to obtain more statistically significant averages over longer time periods without much loss of information (Aubinet et al., 2012). Therefore, the authors did not venture out into testing other input averaging intervals.

References:

Aubinet, Marc, Timo Vesala, and Dario Papale, eds. *Eddy covariance: a practical guide to measurement and data analysis*. Springer Science & Business Media, 2012.

Mahrt L., D. Vickers, J. Howell, J. Højstrup, J. M. Wilczak, J. Edson, and J. Hare, 1996: Sea surface drag coefficients in the Risø Air Sea Experiment. J. Geophys. Res., 101, 14 327–14 335.

Sun, J., J. Howell, S. K. Esbensen, L. Mahrt, C. M. Greb, R. Grossman, and M. A. LeMone, 1996: Scale dependence of air–sea fluxes over the western equatorial Pacific. J. Atmos. Sci., 53, 2997–3012.

Vickers, D., and L. Mahrt, 1997: Quality Control and Flux Sampling Problems for Tower and Aircraft Data. J. Atmos. Oceanic Technol., 14, 512–526.

**P4 L7: Can you elaborate further on how wind speed data is stationary? Is this tested at the prediction timescales (10 min – 3 hr)? I would also expect wind speed to be subject to seasonal and diurnal variability. Please clarify.**

We have used the Augmented Dickey Fuller Test to check for long-term statistical stationarity within a given times series. This test has a null hypothesis that a given time series has a unit root, i.e. that it has a stochastic trend/drift that pervades throughout the entire time series. The testing procedure is applied to the model:

$$\Delta y_t = \alpha + \beta t + \sum_{i=1}^{n} (\delta_i \Delta y_{t-i}) + \varepsilon_t$$

where $\Delta y_t$, in our case, is the change in wind speed from one period to the next, $\alpha \neq 0$ represents a constant drift term, $\beta \neq 0$ represents a trend in the data, $\delta_i$ represents the dependency on the past $\Delta y_{t-i}$ term, and $\varepsilon_t$ is the residual. The number of lags, $n$, is chosen based on the Akaike information criterion (a standard process). The test results in a test statistic (the Dickey-Fuller test statistic) which can be transformed into a p-value that informs the user as to whether or not the null hypothesis (that the time series has a trend/drift) is likely to be true. The goal of this test is determining if the time series has any trend or drift that must be accounted for when running the ARIMA model. Generally speaking, we would like a p-value of $\leq 0.01$ (1% likelihood) to prove that the null hypothesis is false.

We tested for the likelihood that the data (the 10-minute, hourly, and 3-hour time series) could be represented by two basic regression models (these are the models most commonly tested in this type of analysis): a time series with a constant and a trend ($\alpha$ and $\beta \neq 0$) and a time series with a constant and no trend ($\alpha \neq 0$ and $\beta = 0$). Tests of all three time series on both regression models showed a p-value $\ll 0.01$ (the computer-generated p-values were all at least four orders of magnitude smaller than the 0.01 cut-off, meaning there was at most a 0.0001% chance of the null hypothesis being true), providing strong evidence that there is no underlying trend (i.e. change in the mean or variance of the wind speed over the course of the 3+ month campaign) in any of the time series.

To clear up what we believe may be the source of confusion, this test does not take into account any type of diurnal wind speed variations, instead testing to ensure there are no long-term trends/drift in the data. These diurnal variations are expected to be one constituent piece of the ARIMA forecasting error. We have changed the wording to "long-term statistical stationarity" and "wind speed data contains no embedded trends or drift (e.g. changes in the mean or variance of the wind speed due to long-term variability)" (beginning P5 L18) in order to relieve any confusion. We have also added the Python library utilized to perform the tests. However, we would prefer not to include the more detailed analysis above as this test was only one small ancillary piece of the analysis performed.

**P5, L28: I'm curious why are you not using the Obukhov length (or z/L)? Isn't it a more commonly used parameter to characterize stability? You may want to motivate this selection even though from the results of Figure 6 it seems that stability parameters are not that important in the improvement of forecasts.**

We have decided against using the Obukhov length for this study because a few of the theory's critical assumptions (specifically spatial homogeneity) are broken in this location because of the complex terrain. Studies have shown that using the Obukhov length in complex terrain can lead to poor results (e.g., Fernando et al., 2015), and thus we have removed it from the list of potential input features. This has been noted on P7 L10.

References:
Fernando, H. J. S., et al. "The MATERHORN: Unraveling the intricacies of mountain weather." *Bulletin of the American Meteorological Society* 96.11 (2015): 1945-1967.

**P7, L28: How are sonic measurements corrected for tilt? What is the interval used when deriving the fluxes? Is it equal, shorter or longer than the 5-min interval used in the moving average? This is just to know if 5-min is the actual filter in the data or if the data already came with a longer averaging time. This could also be relevant to understand the potential impact of this filter in the performance at 10 min prediction horizon (Figure 3).**

The sonics were corrected for tilt via the technique described in Wilczak, Oncley, and Stage, 2001, "Sonic Anemometer Tilt Correction Algorithms," *Boundary Layer Meteor.,* 99, pp.127-150. This has been noted on P3 L26. The mean values for turbulent flux calculations were taken at 5-minute intervals, so the filter should not have had any effect on the 10-minute prediction forecasts.

**P8, Figure 2: The map is difficult to read. It would be better to show an elevation contour plot where we can read the relative heights. I don't think it is necessary to provide an illustration of the mast levels if they are described in the text.**

Fig. 2 (now Fig. 1) has been replaced with a contour plot with a marker for the tower position

**P8, L8: You end up using 5-min averaged data to build predictive models with prediction horizons at 10 min, 1 hour and 3 hours. You previously mentioned that these are single-step forecasts. Wouldn't you have to use input data that is averaged at the same interval than the forecast step (e.g. use 3-hour moving averages to predict 3 hr ahead)? Or do you forecast {10min, 1hr, 3h} ahead based always based on 5-min data? If the latter is true, please clarify why not using a consistent interval between input and prediction data or, alternatively, how dependent are the results to the chosen interval in the time series.**

The reviewer is correct, the data had already been 5-minute averaged (default from NCAR). We then had to average multiple 5-min periods in order to get the 10-min, hourly, and 3-hour averages for all variables. We have clarified this point on P4 L3.

**P10. Figure 4: One may wonder how a Persistence-RF model would work. This might be a good result to include in the paper so that you can just isolate the impact of RF from that of the forecasting model to make the results more generally applicable. Maybe you get to the same conclusions with a simpler model.**

We agree that this would be an interesting aspect to investigate, but we would rather exclude such tests for a few reasons. First, we worry that adding a Persistence-RF model would cause additional complications around justifying the feature set for such a model and the perceived need of designing another "optimal RF architecture". We would also rather keep the focus of the manuscript on how the random forest benefits a non-naïve model that is a function of previous atmospheric conditions (this idea is further described in the introduction). Additionally, we are currently working on a paper investigating the efficacy of a Persistence-RF model on decadal datasets (FINO1 and ARM SGP), and if readers are curious about such a setup for wind speed forecasting, they may refer to the upcoming manuscript. Therefore, we prefer to refrain from adding results from a Persistence-RF model – no changes.

**P8. L11: How is the flux Richardson calculated between 100-20 m? Isn't it a local quantity derived from a sonic level? Is it the mean value between the two levels? Please clarify.**

In order to calculate the flux Richardson number, we used the $\overline{w'T'}$, $\overline{u'w'}$, $\overline{v'w'}$, and $T_{pv}$ measurements from the 100 m sonic anemometer (this has been noted on P16 L12). However, we used the $u$ and $v$ measurements from both the 100 and 20 m locations in order to calculate $\partial u / \partial z$

and $\partial v / \partial z$. While we recognize that the flux Richardson number is typically more localized, this technique was deemed as the best suited for this particular use case.

Ref: wes-2019-58
Title: Utilizing Physics-Based Input Features within a Machine Learning Model to Predict Wind Speed Forecasting Error
Journal: Wind Energy Science

**Referee:** Anonymous Reviewer

We would like to thank the reviewer for his/her time in reading and commenting on the manuscript that led to considerable improvement of the paper. We have tried to address all comments and hope that this next version is acceptable for publication.

Major Comments:

1. I find the choice of the authors of (extensively) describing the methods used before the data a bit confusing. I personally had to go back to the methods section after reading the data section to make sure I got everything right. I would recommend switching the order of the two sections.
We agree with the reviewer that the order of the sections may lead to confusion/frustration. We have moved the section "Site, Data, & Instrumentation" ahead of the methodology section in order to relieve this issue.

2. More clarification is needed on what averaging time is used in the calculations of the variables considered in this work. For example, what averaging time is used to calculate the Reynolds decomposition for turbulent averaging, for example for TKE, TI, friction velocity? Why did you choose it? How does that conciliate with the different lead times of the ML models?
We have copied a majority of our response from the previous reviewer, as he had a similar question:
The utilized data had already been preprocessed and 5-minute averaged by NCAR; we had forgotten to include this information in the original and have added a reference on P3 L28.
5-minute averaging is a common averaging period used in most meteorological studies (for example CASES-99, RASEX, Perdigão, etc.) as it helps minimize flux sampling errors (systematic, random, and mesoscale variability error) and provides necessary flags to categorically distinguish between instrumental problems and plausible physical behavior (Mahrt et al. 1996, Sun et al. 1996, Vickers and Mahrt 1997). A local average of 5 minutes seems to adequately capture most of the turbulent fluxes in stationary time periods compared to one-hour local averaging (Mahrt et al., 1996, Sun et al., 1996). A 20 to 30-minute time-averaging protocol has become standard eddy-covariance practice for idealized conditions (i.e., quasi-stationary and horizontally homogeneous), but one can combine these 5-minute averages to obtain more statistically significant averages over longer time periods without much loss of information (Aubinet et al., 2012). Therefore, the authors did not venture out into testing other input averaging intervals.

Figure 2a has been replaced with a contour plot (Fig. 1) with a marker for the tower position

7. Figure 2-b: not really needed.
Figure 2b has been removed from the manuscript

8. P. 4: was wind speed at Perdigão really stationary? Over which time scales? Please clarify.
We have copied our response from the previous reviewer, as he had a similar question:

We have used the Augmented Dickey Fuller Test to test for long-term statistical stationarity within a given times series. This test has a null hypothesis that a given time series has a unit root, i.e. that it has a stochastic trend/drift that pervades throughout the entire time series. The testing procedure is applied to the model:

$$\Delta y_t = \alpha + \beta t + \sum_{i=1}^{n}(\delta_i \Delta y_{t-i}) + \varepsilon_t$$

where $\Delta y_t$, in our case, is the change in wind speed from one period to the next, $\alpha \neq 0$ represents a constant drift term, $\beta \neq 0$ represents a trend in the data, $\delta_i$ represents the dependency on the past $\Delta y_{t-i}$ term, and $\varepsilon_t$ is the residual. The number of lags, $n$, is chosen based on the Akaike information criterion (a standard process). The test results in a test statistic (the Dickey-Fuller test statistic) which can be transformed into a p-value that informs the user as to whether or not the null hypothesis (that the time series has a trend/drift) is likely to be true. The goal of this test is determining if the time series has any trend or drift that must be accounted for when running the ARIMA model. Generally speaking, we would like a p-value of $\leq 0.01$ (1% likelihood) to prove that the null hypothesis is false.

We tested for the likelihood that the data (the 10-minute, hourly, and 3-hour time series) could be represented by two basic regression models (these are the models most commonly tested in this type of analysis): a time series with a constant and a trend ($\alpha$ and $\beta \neq 0$) and a time series with a constant and no trend ($\alpha \neq 0$ and $\beta = 0$). Tests of all three time series on both regression models showed a p-value $\ll 0.01$ (the computer-generated p-values were all at least four orders of magnitude smaller than the 0.01 cut-off, meaning there was at most a 0.0001% chance of the null hypothesis being true), providing strong evidence that there is no underlying trend (i.e. change in the mean or variance of the wind speed over the course of the 3+ month campaign) in any of the time series.

To clear up what we believe may be the source of confusion, this test does not take into account any type of diurnal wind speed variations, instead testing to ensure there are no long-term trends/drift in the data. These diurnal variations are expected to be one constituent piece of the ARIMA forecasting error. We have changed the wording to "long-term statistical stationarity" and "wind speed data contains no embedded trends or drift (e.g. changes in the mean or variance of the wind speed due to long-term variability)" (beginning P5 L18) in order to relieve any confusion. We have also added the Python library utilized to perform the tests. However, we would prefer not to include the more detailed analysis above as this test was only one small ancillary piece of the analysis performed.

9. P. 6 L. 10: Rephrase as "a feature set that utilizes all input features is tested"
This phrasing has been added to the manuscript (P8 L9).

10. Did you apply any cross-validation for your ML models? If not, why?
By cross-validation we assume the reviewer is referring to the use of a validation set during the training process. We did not use a validation set. Unlike an artificial neural network, the random forest model does not require a validation set as it is inherently robust against the problem of overfitting (Breiman, 2001). The bagging process, when combined with a large number of trees and effective pruning (all described in Sec. 3.2), effectively obviates the necessity of a validation set – no changes.

11. P. 8 L. 1: Please specify what you mean by "sensors at 20 and 100 m AGL were chosen based on data availability."
These sensors were chosen because they had a relatively high percentage of clean data. We have clarified this statement: "the high percentage (>99% for all variables except temperature at 100 m AGL, which was available for ~95% of the periods) of clean data at these elevations" (P3 L30).

12. Please state the native time resolution of the sonic data you are using.
A statement has been added to P3 L25: "(20 Hz native measurement resolution)".

13. Have sonic anemometer data been filtered for tower wake effects? These effects would artificially increase turbulence (and reduce wind speed) for some wind direction bins, thus invalidating the quality of quite some data.
A line has been added to P3 L27 stating that no clear tower wake effects could be discerned. During quality control, the data has been checked for tower wake effects, but the two primary effects cited in the literature (reduced wind speeds alongside increased TKE; Barthlott and Fiedler 2003; McCaffrey et al. 2017) were not discerned in the dataset. The boom was angled at ~135º from northerly, meaning that the center of the tower wake would be expected at ~315º, approximately parallel to the ridge. Both the average wind speed and TKE in the expected wake region were similar to that seen in the opposite direction (135º), as can be seen in the figure below (dashed vertical lines indicate directions of along-ridge flow; 135º is opposite to expected wake, 315º is expected wake region; data is separated into 5º bins). Because we did not perceive wake effects, no corrections were made – no changes.

[Figure]

References:
Barthlott, Christian, and Franz Fiedler. "Turbulence structure in the wake region of a meteorological tower." *Boundary-layer meteorology* 108.1 (2003): 175-190.

[revised manuscript text omitted]

---

## Referee Report (RR1)

**Utilizing Physics-Based Input Features within a Machine Learning Model to Predict Wind Speed Forecasting Error**

*Daniel Vassallo, Raghavendra Krishnamurthy and Harindra J.S. Fernando*

**REVIEW – round 2**

**GENERAL COMMENT:**

The authors have addressed most of my comments from the previous review, and the manuscript has improved. There are, however, a couple of major points that I think still need to be tested/addressed before the manuscript can be recommended for publication.

**MAJOR COMMENTS:**

- Page 3 line 28: In response to my previous comment on the topic, you now state that "No clear tower wake effects could be discerned." However, I do not think you have performed a necessarily correct check. The fact that wind speed and TKE are similar in magnitude for opposite wind directions does not necessarily mean that tower wake effects are not present: why should wind speed and TKE be equal in the first place for opposite wind directions? The correct way to assess potential impacts of tower wake effects would be to compare concurrent wind speed and TKE values as a function of wind direction as measured by two sonic anemometers mounted on opposite booms on the same met tower at the same height. From my knowledge, this configuration can be found in one of the meteorological towers at Perdigão. I strongly advise the authors to re-assess this aspect.

- Page 4: Given the randomized splitting between training and testing datasets, together with the absence of cross-validation, I am still concerned about potential overfitting, also considering the large autocorrelation your data have (due to the sum of that introduced by the overlapping averages and that naturally present in the data). While you state that random forests do not overfit the data at all, this is a debatable statement: I am sure you can find plenty of papers that can support both opinions. To make this reviewer happy, I would love to see whether the performance of the proposed model varies if the splitting between training and testing set is not performed randomly, but rather with a hybrid approach. For example, what happens if you keep all observations from one week for testing? Or from one full day from each week? Such tests would definitely give an answer to potential autocorrelation impacts on your results.

**MINOR COMMENTS:**

- Figure 1: this map still looks somewhat incomplete to me: at the very least, please add some reference to understand the horizontal distances.

- Page 4 l. 3: what do you mean by 'augmented' data here?

- Page 4 l.3: "data were averaged into 10-minute, hourly, and three-hour segments at a 5-minute moving average in order to create a robust dataset" is still not clear to me. Do you mean that you are creating three datasets, both with one data point every 5-minute, but in dataset A all data are 10-minute average, in dataset B hourly averages, and in dataset C 3-hourly averages?

- Figure 5: these scatterplots could be improved. Can you please change the color in the scatterplot based on density (e.g. https://matplotlib.org/api/_as_gen/matplotlib.pyplot.hist2d.html)?

---

## Author Response (AR2)

Ref: wes-2019-58
Title: Utilizing Physics-Based Input Features within a Machine Learning Model to Predict Wind Speed Forecasting Error
Journal: Wind Energy Science

**Referee:** Anonymous Reviewer

We want to sincerely thank the reviewer for once again taking the time to provide an in-depth review of the paper. The comments have helped us make considerable improvements to the manuscript and we hope the updated version is acceptable for publication.

Major Comments:

1. Page 3 Line 28: In response to my previous comment on the topic, you now state that "No clear tower wake effects could be discerned." However, I do not think you have performed a necessarily correct check. The fact that wind speed and TKE are similar in magnitude for opposite wind directions does not necessarily mean that tower wake effects are not present: why should wind speed and TKE be equal in the first place for opposite wind directions? The correct way to assess potential impacts of tower wake effects would be to compare concurrent wind speed and TKE values as a function of wind direction as measured by two sonic anemometers mounted on opposite booms on the same met tower at the same height. From my knowledge, this configuration can be found in one of the meteorological towers at Perdigão. I strongly advise the authors to re-assess this aspect.

      We were unaware that there were two sonic anemometers at the same height on a single tower, and we have since performed the suggested assessment. The test showed small shadow effects (maximum of 7% decrease in wind speed, shown below) in a directional sector spanning about 30º (~310-340º from northerly), a much smaller shading effect than we have seen quoted in much of the literature, which are generally more than 30% (e.g. Moses & Daubek, 1961; Cermak & Horn, 1968; Orlando et al., 2011; McCaffrey et al., 2017, Lubitz et al., 2018). After examining this effect, we have added a statement in the manuscript stating that small tower shading effects were present for this sector (beginning P3 L26). However, we would still prefer to keep the unaltered data in the study for two reasons.

      First, the ARIMA model is particularly useful for continuous datasets and has a built-in assumption that the dataset is continuous. As mentioned in Section 2, we have filled in missing data periods because we wanted to ensure that the model would have a continuous dataset. The 310-340º directional sector constitutes approximately 5% of the dataset and removing this data would lead to considerable data partitioning, thereby negating much of the efficacy of the ARIMA model. Further, all periods which utilize the removed data as inputs would themselves be negated. The dataset is already relatively small for a machine learning application, and we would therefore prefer to avoid removing these periods, especially due to the finding that the shading effect is on the order of 7%.

      We would also prefer not to replace the data via a correction such as the measure-correlate-predict (MCP) method utilized to fill the missing periods (Section 2). While we were required to use the method to fill missing periods, it can produce errors (~17% mean absolute error) that are greater than that seen from the tower shadow effect. We do not expect that this would be a problem

for the filled periods, as they constitute only a very small portion of the dataset (less than 1%). However, filling 7-8% of the dataset via MCP could lead to misleading results.

We would therefore prefer to keep the dataset as-is, as any adjustment would likely lead to further complications and potentially deleterious results. As stated previously, we have added a statement in the manuscript that slight tower shading was observed, but we have left the data unaltered as it was far less prominent than that quoted in the literature. We would like to thank the reviewer for referring to the information that led to this finding.

We appreciate this comment. In reviewing the cross-validation, we indeed discovered that the model was overfitting the data. We have made appropriate changes to the manuscript to reflect the new findings. Much of the results and discussion (which are now a single section to ease the explanation process) have been changed to reflect the changes in findings. The new findings reflect the results obtained over 10 testing sets partitioned via stratified k-fold cross validation. Similar to the reviewer's suggestion, this cross-validation technique splits the data nearly chronologically while ensuring the target variable distribution is consistent among the training and testing sets. We would like to thank the reviewer for his/her suggestion of performing cross validation, as it has led to many positive updates to the manuscript.

Minor Comments:

Figure 1: this map still looks somewhat incomplete to me: at the very least, please add some reference to understand the horizontal distances.
We have added both a reference for horizontal distances as well as a north arrow to make it easier for readers to orient themselves.

Page 4 L. 3: what do you mean by 'augmented' data here?
Our intention was to mention that the data has been quality controlled, and we have changed "augmented" to "quality controlled" to better reflect this (P4 L7).

Page 4 L. 3: "data were averaged into 10-minute, hourly, and three-hour segments at a 5-minute moving average in order to create a robust dataset" is still not clear to me. Do you mean that you are creating three datasets, both with one data point every 5-minute, but in dataset A all data are 10-minute average, in dataset B hourly averages, and in dataset C 3-hourly averages?
Yes, the reviewer is correct. We have changed the language of the sentence to "data were averaged over 10-minute, hourly, and three-hour segments at a 5-minute moving average in order to create three robust datasets, each consisting of over 28,000 samples" in order to relieve any potential confusion (P4 L7).

Figure 5: these scatterplots could be improved. Can you please change the color in the scatterplot based on density (e.g. https://matplotlib.org/api/_as_gem/matplotlib.pyplot.hist2d.html)?
We have removed this figure as it is no longer a useful indicator of model performance. We have replaced the $R^2$ metric with mean absolute error (MAE) as we believe it is a more telling metric.

[revised manuscript text omitted]